# Experimental Investigation into the Seismic Performance of Fully Grouted Concrete Masonry Walls Using New Prestressing Technology

**Bin Chi [1], Xu Yang [1], Fenglai Wang [2,3,\*], Zhiming Zhang [1] and Yuhu Quan [1]**

[1] School of Civil Engineering, Harbin Institute of Technology, Harbin 150090, China;
binchi.hit@hotmail.com (B.C.); yangxu_phdce@mail.com (X.Y.); zhangzhiming724@hotmail.com (Z.Z.);
quanyuhuhit@outlook.com (Y.Q.)

[2] Key Lab of Structures Dynamic Behavior and Control of the Ministry of Education, Harbin Institute of
Technology, Harbin 150090, China

[3] Key Lab Smart Prevention and Mitigation of Civil Engineering Disasters of the Ministry of Industry and
Information Technology, Harbin Institute of Technology, Harbin 150090, China

\* Correspondence: fl-wang@hit.edu.cn; Tel.: +86-139-3616-8890

**Abstract:** In recent years, traditional masonry structures have been widely used in rural areas of China. However, they were found to have a poor seismic performance during earthquakes. In this study, a new prestressing technology was proposed and described in detail, and it was used in fully grouted concrete masonry wall systems to improve its seismic performance. The experimental work involved investigating the seismic response of four fully grouted reinforced concrete masonry wall systems, consisting of two symmetrically arranged reinforced block masonry walls, with different section types and prestressing technologies, when subjected to cyclic lateral force. Based on the test results, a flexure and ductile failure occurred in the specimens with a rectangular section, while a shear and brittle failure occurred in the specimens with a T-shape section. The prestressing technology had no significant effect on the failure state of the specimens, but it influenced the crack propagation, making cracks fine and densely covered. A symmetrical and obvious pinching effect was observed in the hysteretic response of all specimens. The average displacement ductility of the specimens varied within a range of values between 3.34 and 6.92, according to the section type of the specimens, and the prestressing technology improved the displacement ductility of the specimens. Moreover, the prestressing technology significantly improved the initial stiffness of the specimens, and the specimens with prestressing technology experienced a greater fall in the degradation of the normalized stiffness than the specimens without this technology throughout the loading process. In addition, the equivalent viscous damping of the specimens ranged between 8.2% and 10.8%, according to the section type. It could be concluded that the prestressing technology improved the energy dissipation of the specimens at the ascending stage, although it had no marked influence on the equivalent damping ratio of the specimens.

**Keywords:** reinforced masonry; prestressing technology; experimental investigation; cyclic loading; seismic performance

## 1. Introduction

Masonry structures are considered as a traditional structural form widely used in the world for many years. While reinforced concrete structures have become dominant in urban construction due to their fast construction speed in recent decades, masonry structures are still considered the preferred option in rural construction in China because of the simplicity and inexpensiveness of masonry

materials. However, masonry structures were observed to have a poor seismic performance, and 74% of all brick masonry buildings collapsed or were seriously damaged in the Wenchuan earthquake [1]. As a result, improving the seismic performance of masonry structures in rural zones has become a research hotspot.

There are numerous experimental and theoretical studies on various aspects of masonry structures, with the aim of improving their seismic performance. On the one hand, research on new construction forms has been carried out. Haach et al. [2] proposed an innovative horizontal trussed reinforcement for reinforced concrete masonry walls, and the influence of the horizontal reinforcement on the seismic performance of the specimens was studied using cyclic tests. The results stressed that the presence of the horizontal reinforcement ensures a better control and better distribution of cracking. Voon and Ingham [3] studied ten single-story reinforced concrete masonry walls and the influence of the applied axial compressive stress and shear reinforcement on the seismic performance of the specimens. The results confirmed that the shear strength of the specimens increases the magnitude of the applied axial stress. Similarly, research on seismic performance of reinforced concrete masonry walls also conducted by EI-Dakhakhni et al. [4], Bolhassani et al. [5], Ramirez et al. [6], Eldin et al. [7–10], and Obaidat et al. [11]. Moreover, Ma et al. [12] used a special type of core-aligned block (double H-block) to construct a shear wall, and cyclic load testing was carried out. The results showed that the double H-block masonry shear wall exhibited a high ductility and energy dissipation. Wang et al. [1] designed a new type of precast concrete interlocking block and used it as fabricated columns, and comparison experiments were carried out to evaluate the seismic performance of masonry walls with traditional columns. The results demonstrated that the implementation of fabricated columns improved the seismic response with respect to traditional construction practices. On the other hand, some commonly used strengthening technologies are adopted in new structures to improve their seismic performance. One of the technologies is the use of high-strength materials, and a lot of research is already in progress on this topic [13–15]. Zoppo et al. [13] proposed an innovative fiber-reinforced cementitious mortar (FRCM) system to improve the shear capacity of solid clay brick masonry walls, replacing fiber-reinforced mortar (FRM). An experimental program on masonry panels indicates that panels reinforced with symmetric FRM achieved a similar effectiveness in the shear strength increase to that of panels reinforced with symmetrical FRCM, but a reduced deformability is observed in FRM panels, compared with that observed in FRCM. Another method, called prestressing technology, strengthens masonry panels by increasing the axial stress, and an unbounded prestressing technology has recently become the most popular choice. Laursan and Ingham [16] tested six fully grouted, one partially grouted and one ungrouted wall using unbounded post-tensioning. The tests results show that the fully grouted walls exhibited a large lateral drift and small residual deformation, while the partially grouted and ungrouted walls exhibited shear failures. Hassanili et al. [17,18] carried out an experimental study on four unbounded posttensioned masonry walls. The results show that the lateral strength of the specimen is influenced by the distribution of posttensioned bars, with the same total initial posttensioned force. The unbounded post-tensioning specimens are like rocking walls, and similar and relevant research was also carried out by Roseboom and Kowalsky [19], Wight et al. [20], Kalliontzis and Schultz [21] and Ryu et al. [22]. Moreover, Guo et al. [23] proposed a novel concrete masonry wall, called an ungrouted posttensioned confined concrete masonry wall with unbounded tendons, and the masonry panel was posttensioned and confined by the ring beam and constructional columns. The tests results indicate that the post-tensioning is significantly effective in improving the cracking resistance and seismic behavior of the walls. Wight et al. [24] described and discussed previous post-tensioned concrete masonry research applications and post-tensioning construction details. Ma et al. [25] applied the external prestressing technique on a masonry structure, and shaking table model tests were conducted. The experimental results show that prestressing influences the failure mode of masonry structures and improves their torsional resistance.

Therefore, the above studies demonstrate that prestressing is a promising technique to improve the seismic performance of masonry structures. However, the tedious construction method associated

with the unbounded technology and tension technology limits the application of the prestressing technology. Rosenboom and Kowalsky [19] carried out experiments with bonded posttensioned masonry walls but found that the deformation capacity and strength of the specimens was reduced. The reason for this is the bond stress between the posttension duct and surrounding grout. Thus, the influence of the posttension duct cannot be ignored. Furthermore, studies on the seismic performance of reinforced concrete masonry structures concentrate on single-story walls [26–29], and little research has been carried out on collaborative work between two walls. Mavros et al. [30] discussed shaking table tests on a full scale, two-story, reinforced masonry shear-wall structure, with two T-sectioned and one rectangular-section wall components. The test results show that the walls in the orthogonal direction had a significant contribution to the lateral load resistance of the structure. Siyam et al. [31], Heerema et al. [32] and Ashour et al. [33] carried out a series of studies on the inelastic seismic response of reinforced masonry walls at the system-level by cyclic tests on individual components and within two asymmetrical building systems. The results show that the variation in the inelastic response of reinforced masonry walls comprising the buildings, and the interaction relationship between the walls and diaphragm cannot be neglected.

In summary, there are still many problems and unknowns in the application of prestressing masonry structures. Based on the above, a new feasible prestressing technology has been proposed. To accomplish this, one end of vertical bar was machined by threads, and the other end was anchored in the beam or floor. Using the vertical hole in the hollow masonry wall, the vertical bar was fixed at the top of hollow masonry wall by anchorage plate and the prestressing force was applied by torque wrench. Therefore, in this study, the coupled masonry shear walls are selected as the research object, and the influence of prestressing technology on the seismic performance of reinforced concrete masonry walls was investigated using a cyclic test. Four comparison specimens with different section types and prestressing technologies were built, and a complete description of the failure mode and crack pattern of the specimens is presented. Seismic performance indexes, like the displacement ductility, stiffness degradation, energy dissipation and equivalent viscous damping, are discussed.

## 2. Experimental Program

### 2.1. Test Specimens

In this study, the seismic performance of four single-story, single-bay fully grouted reinforced concrete block masonry wall systems, subjected to in-plane cyclic lateral force, were studied, and they consisted of two symmetrically arranged reinforced block masonry walls. In China, infill masonry walls are often formed into a tooth shape at the boundary to enhance the connection with the concrete column [1]. Considering the need for reinforced concrete block masonry walls and infill wall connections, the specimens were designed with the tooth shape at the boundary to keep up with the structural form used in engineering practice. Figure 1 shows the configuration details of all specimens. All specimens consisted of a heavy bottom beam and a top beam to ensure fixed conditions at the bottom and lateral force exertion at the top, respectively. There are two categories in the experiment program: the rectangular section category and T-shape section category, and each category included two comparison specimens, with and without using prestressing technology. Table 1 summarized the dimensions and reinforcement details of the specimens. The abbreviated name of bare coupled masonry walls with rectangular section was BMF, and the abbreviated name of bare couple masonry walls with T-shape section was BMFT. Furthermore, to distinguish the comparison specimens, the specimen BMF with prestressing technology was defined as BMFP, while the specimen BMFT with prestressing technology was defined as BMFTP. Specifically, the reinforced concrete block masonry wall in the rectangular section category (BMF and BMFP) had dimensions of 590 mm × 190 mm × 1600 mm (length × width × height) and uniformly distributed two horizontal plain bars, with a diameter of 6 mm and spacing of 400 mm. The vertical reinforcement consisted of deformable bars, with a diameter of 16 mm, which was arranged at each cell of the specimen. The difference between the BMF and

BMFP was that the vertical reinforcement of the BMFP was machined by threads. Compared to the rectangular section category, the T-shape section category (BMFT and BMFTP) had a flange at each of the reinforced block masonry walls. All reinforced block masonry walls had spiral stirrups on the bottom to enhance the compression performance of the end part of the reinforced block masonry walls.

**Table 1.** Design detail of the specimens.

| Specimen | Load-Bearing | | | | Reinforcement Configuration | | | Prestressing |
|---|---|---|---|---|---|---|---|---|
| | Length (mm) | Width (mm) | Height (mm) | Flange (mm) | Vertical | Horizontal | Spiral | |
| **BMF** | 590 | 190 | 1600 | 0 | 6C16 | 2A6@400 | A6@70 | × |
| BMFP | 590 | 190 | 1600 | 0 | 6C16 | 2A6@400 | A6@70 | √ |
| BMFT | 590 | 190 | 1600 | 590 | 4C14+6C16 | 2A6@400 | A6@70 | × |
| BMFTP | 590 | 190 | 1600 | 590 | 4C14+6C16 | 2A6@400 | A6@70 | √ |

The abbreviated name of bare coupled masonry walls with rectangular section was BMF, and the abbreviated name of bare couple masonry walls with T-shape section was BMFT. Furthermore, to distinguish the comparison specimens, the specimen BMF with prestressing technology was defined as BMFP, while the specimen BMFT with prestressing technology was defined as BMFTP.

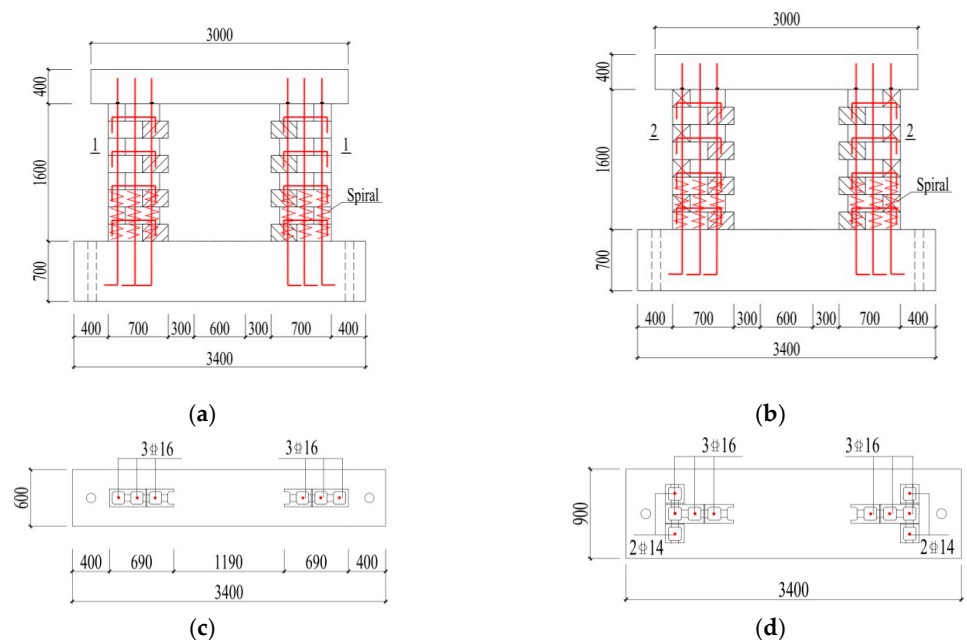

**Figure 1.** Design details of the specimens: (**a**) details of the specimens, BMF and BMFP; (**b**) details of the specimens, BMFT and BMFTP; (**c**) corss section of specimens, BMF and BMFP; (**d**) cross section of specimens, BMFT and BMFTP. The abbreviated name of bare coupled masonry walls with rectangular section was BMF, and the abbreviated name of bare couple masonry walls with T-shape section was BMFT. Furthermore, to distinguish the comparison specimens, the specimen BMF with prestressing technology was defined as BMFP, while the specimen BMFT with prestressing technology was defined as BMFTP.

There were two aspects of the prestressing technology: the magnitude of the prestressing and the application method of the prestressing. Considering the simplicity of the construction and the effectiveness of the prestressing technology, the total prestressing stress selected to be applied to the specimens was 0.453 MPa, which was equal to the design axial compression ratio of 0.1, calculated by the Chinese code [34]. To ensure that the action point of the prestressing force was located at the centroid of section of specimen, the position and magnitude of stress on the bars were determined

by mechanics calculation. For the rectangular section specimen BMFP, the two ends of the vertical reinforcement were selected to apply tension to the prestressing on the masonry wall, and the stress of each vertical bar was 126.4 MPa. For the T-shape section specimen BMFTP, three vertical bars at the end of the specimen were selected to apply tension. The stress on the bars with a diameter of 14 mm was 193.7 MPa, while the stress on the bars with a diameter of 16 mm was 126.9 MPa. The shadow zone in Figure 2a,b represent the position of the prestressing force on the specimens with different sections. Moreover, unlike the previous unbounded prestressing technology, a feasible method for employing the prestressing is shown in Figure 2d. Before concrete masonry wall grouting, the vertical reinforcement was already arranged at each cell of the concrete masonry wall. The lower end of the vertical bar was rigidly anchored to the bottom beam, and the other end with threads exposed at the top of the concrete masonry wall. During the total prestressing construction process, the prestressing forces on the vertical bars were applied by a torque wrench and controlled by a preset torque. The vertical bars were tensioned and fixed at the top of the hollow masonry wall by anchorage plates and screw nuts (see Figure 2d) in the process of concrete grouting and vibration.

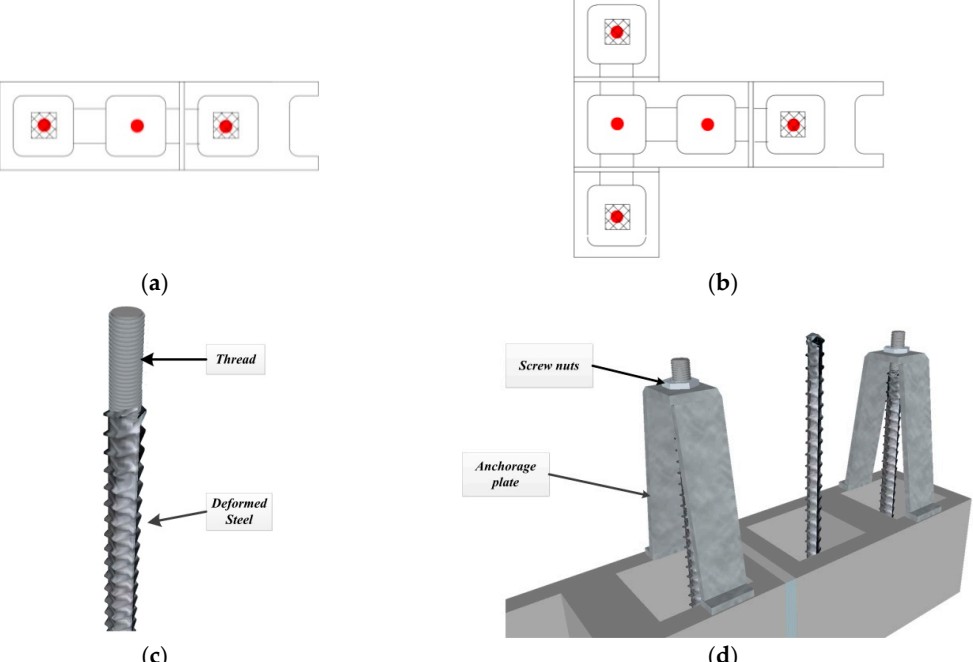

**Figure 2.** Details of the prestressing technology: (**a**) the prestressing position of the BMFP specimen; (**b**) the prestressing position of the BMFPT specimen; (**c**) steel with thread; (**d**) anchorage type.

### 2.2. Wall Construction

All specimens were constructed by experienced masons using a running pattern, and the precast technology was adopted. The construction process is shown in Figure 3. The hollow masonry walls and bottom beams were built up in adjacent locations at the same time. The horizontal reinforcement was placed on the hollow masonry walls, and the vertical reinforcement was resettled at the bottom beams in advance. After the hollow masonry walls were cured, they were fastened and placed on the bottom beam by lifting technology and adjusted vertically on the bottom beam by a steel strut, which was anchored to the ground. After that, the interfaces between the hollow masonry walls and bottom beams were filled with mortar. During the grouting process, the prestressing technology was applied as described in the previous section. The top beams of all specimens were constructed at the same time, and the specimens were cured under the appropriate environmental conditions.

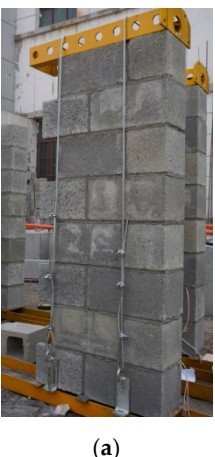
(**a**)
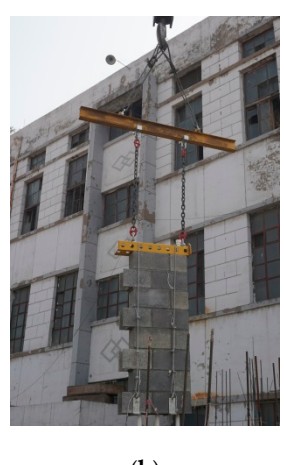
(**b**)
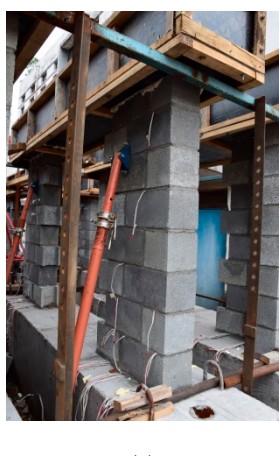
(**c**)

**Figure 3.** Construction of the specimens: (**a**) view of wall prefabrication; (**b**) view of wall lifting; (**c**) view of wall positioning.

*2.3. Material Properties*

To obtain material information on the specimens, the sample had been reserved in the process of specimen construction. The mortar and concrete sample had dimensions of 70.7 mm × 70.7 mm × 70.7 mm and 100 mm × 100 mm × 100 mm, respectively, in accordance with the Chinese regulations [35]. Moreover, considering the complex combination of masonry walls, six masonry prisms were constructed and cured at the same time as the specimens, according to the Chinese standard [36]. Additionally, the deformable bars and plain bars were used as vertical reinforcements and horizontal reinforcements, and the sample was also reserved for testing its tensile strength. Table 2 summaries the test results of the all materials.

**Table 2.** Mechanical properties of materials.

| Material. | Characteristic | Symbol | Strength Value | Coefficient of Variation |
|---|---|---|---|---|
| Mortar | Compressive strength | $f_{m,mortar}$ | 11.7 MPa | 6.2% |
| Concrete | Compressive strength | $f_{m,concrete}$ | 29.8 MPa | 4.5% |
| Masonry prism | Compressive strength | $f_{m,prism}$ | 18.4 MPa | 1.7% |
| | Modulus of elasticity | $E_{m,prism}$ | $1.6 \times 10^5$ N/mm$^2$ | 2.1% |
| Deformable bar | Yield strength | $f_{yv}$ | 458.4 MPa | 1.0% |
| | Ultimate strength | $f_{uv}$ | 648.2 MPa | 0.8% |
| Plain bar | Yield strength | $f_{yh}$ | 226.5 MPa | 7.0% |
| | Ultimate strength | $f_{uv}$ | 255.4 MPa | 8.9% |

*2.4. Testing Methodology*

2.4.1. Test Setup and Instrumentation

Figure 4 shows the test setup of all specimens. Because the masonry wall system considered in this study can be used in single-story buildings with a wooden roof, the axial load on the walls would not be too large. To accommodate the practical conditions and study the influence of the prestressing technology, the specimen was anchored to the lab floor, and no axial stress was imposed on the specimen. The lateral force exerted by the MTS hydraulic actuator, with a capacity of 1000 kN, was installed at the reaction wall at a height of 2500 mm. The MTS hydraulic actuator and top beam of specimen were connected by spherical hinge, which were to ensure the lateral force acting in the center of the top beam during the loading process. Moreover, two groups of parallel lateral bracings were used at both sides of the specimens to avoid out-of-plane displacement. Moreover, the instrumentations of the specimens were also plotted in Figure 4. The linear variable displacement transducers (LVDTs) were placed on the one side of specimen. LVDT 1 was used to measure the lateral displacement of

the top beam of the specimen, while the LVDT 2 was located at the center line of the bottom beam to measure the slip displacement of the bottom beam. The control displacement was measured as the difference between the measured values of LVDT 1 and LVDT 2. LVDT 3 was arranged in the vertical direction of the bottom beam to monitor the lift of the bottom beam in real-time.

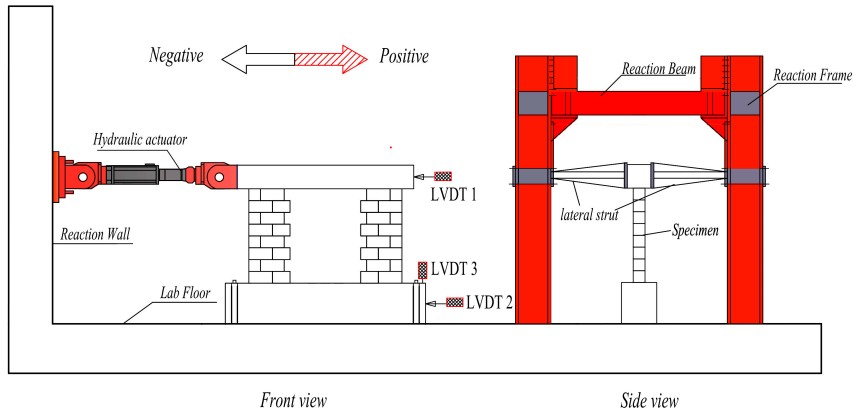

**Figure 4.** Test setup.

### 2.4.2. Load Sequence

Considering the cracking characteristics of the masonry materials and conforming to the Chinese regulations [37], the load and displacement hybrid loading mode was adopted in this study, which consisted of a load control and displacement control. Figure 5 shows the loading sequence, which was also used in [27–29]. At the first, the specimen was controlled by a load, and the loading cycle was repeated once. Once the inflection point occurred at the load–displacement curves, the displacement control was adopted to replace the load control, and two cycles were completed for each target displacement increment. In order to fully obtain the seismic performance data of the specimens in the large displacement, and considering the safety of personnel, the test stopped when the lateral resistance dropped to 60% of the maximum lateral load recorded in the loading process, which was lower than the test stop condition recommended in literatures [6–10].

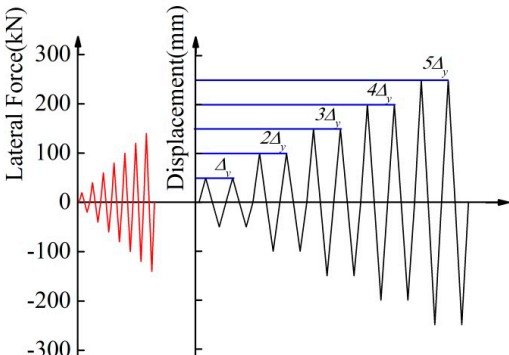

**Figure 5.** Schematic diagram of load sequence.

## 3. Test Observations and Results

### 3.1. Global Response and Crack Pattern

Figure 6 showed the final end state of all specimens. Two kinds of failure modes were observed. For the specimens with a rectangular section (BMF and BMFP), the cracks were either stepped cracks or diagonal cracks, and the cracks were concentrated at the top and bottom of the specimens, while few cracks appeared in the middle. The stepped cracks passed through the interface between the masonry units and mortar, while the diagonal cracks mainly appeared at the face of the masonry units.

The stepped cracks were caused by a high shear stress, which exceeded the bond strength between the masonry units and mortar, while the diagonal cracks were due to the principal stress on the face being larger than the tensile strength of the masonry units. Moreover, the face of the masonry at the bottom of the wall was broken, and grouted concrete was exposed. This mainly related to the complex stress state of the repeated tension and compression on this zone due to cyclic loading. With regard to the specimens with a T-shape section (BMFT and BMFTP), the cracks were also of two kinds, stepped cracks and diagonal cracks, and the cracks developed exclusively on the surface of the masonry wall. The cracks firstly appeared at the interface between the masonry units and mortar located at the bottom of the masonry wall, and gradually developed from both ends to the middle of the masonry wall, which did not occur in the specimen with a rectangular section. A crossing crack occurred at the middle of the masonry wall, when the specimen reached its maximum capacity, and masonry face spalling occurred in that zone due to the repeated cyclic test. Additionally, only bed-joint cracks occurred at the flange, and there was no face spalling at the bottom of the flange (Figure 6e,f). This appearance can be explained by the influence of the flange, which enhanced the compression performance of the masonry wall. Furthermore, a main vertical crack occurred at the web face of the masonry wall, and its width was larger at the height of 400 mm or 800 mm than it was at a height of 600 mm, as shown in Figure 6c,d. This indicated that the crack propagation was related to the horizontal reinforcement, and the horizontal reinforcement diminished the crack width, which was arranged at the height of 600 mm.

With regard to the prestressing technology, the failure mode and crack propagation of the specimens with the same section type were similar. However, the cracks in the specimen BMFTP were thin and densely covered, while the cracks in the specimen BMFT formed several wide cracks. This indicated that the prestressing technology had no significant effect on the failure state of the specimens but influenced the crack propagation. However, there needs to be more studies to prove this.

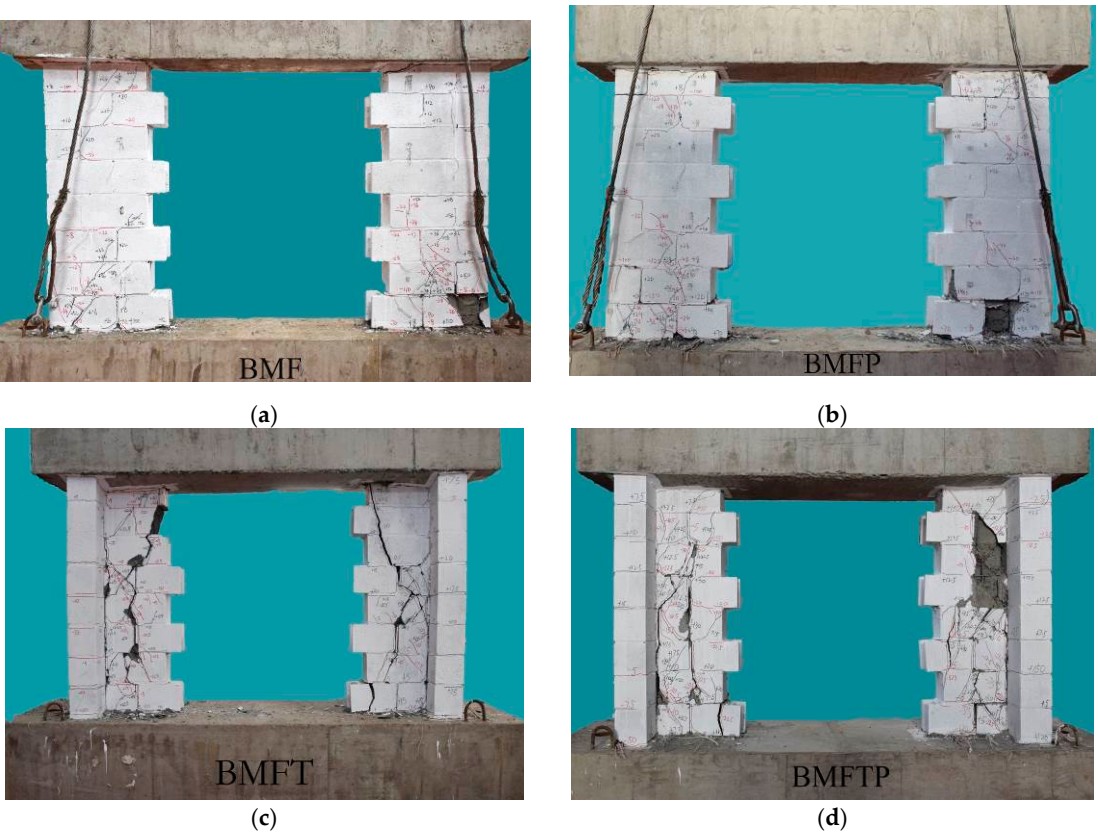

**Figure 6.** *Cont.*

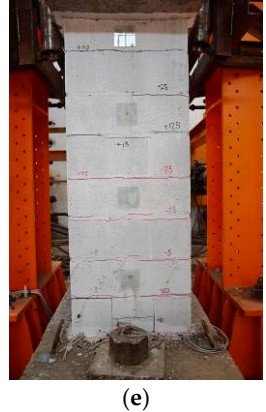

(**e**)

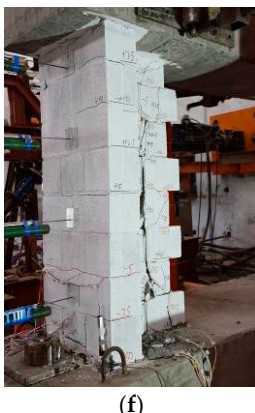

(**f**)

**Figure 6.** Failure modes of the specimens: (**a**) BMF, (**b**) BMFP, (**c**) BMFT, (**d**) BMFTP, (**e**) flange view of BMFT, (**f**) flange view of BMFTP.

### 3.2. Load–Displacement Response

Figure 7 presents the load–displacement response for each specimen, and the corresponding envelope curve is also plotted in each figure using red lines. As for the drift ratio (DR) related to the actual height of the lateral force, 1800 mm was selected due to the distance between the lateral actuator and top face of bottom beam. Therefore, the drift ratio can be defined as the ratio of the displacement and actual height of the lateral force, and it is also plotted in the top axis of the load–displacement response. Moreover, four special states are also identified in the figure: the crack state, max state, ultimate state and final state. The crack state represents the first crack that occurred in the specimens, and the max state indicates that the specimen reached its maximum lateral resistance in the positive or negative direction. As for the ultimate state, there were two definitions commonly used in the literature [6–10,27–29], but it can be considered the state in which the lateral resistance of the specimen decreased to 85% of the maximum lateral resistance, as recommended by the Chinese Code [37]. The final state is the test stop state, when the lateral resistance of the specimen dropped to 60% of the maximum lateral resistance in the positive or negative direction. The measured displacement and load of each specimen were collected, as shown in Table 3. The terms Q and △ represent the lateral resistance and corresponding displacement of specimens, respectively.

In general, all specimens showed a symmetrical hysteretic response and obvious pinching effect. The load–displacement curves for specimens with a different section type showed a different downward trend. The lateral resistance of the T-shape section specimens decreased more abruptly than that of the rectangular section specimens. This can be explained by the fact that a flexure and ductility failure occurred in the rectangular section specimens, while a shear and brittle failure occurred in the T-shape section specimens. Additionally, for specimens BMF and BMFP, the load–displacement curves are similar, but the descending section of the curves is gentler in BMFP than it is in BMF, which is shown by the spacing between the max state and ultimate state in Figure 7a,d. A similar situation also occurred in specimens BMFT and BMFTP. With regard to the measured load and displacement, listed in Table 3, the lateral resistance of the specimen BMFP is 7% higher than that of the specimen BMF, while the lateral resistance of the specimen BMFTP is close to that of the specimen BMFT. Moreover, the specimens with or without prestressing technology reached their maximum lateral resistance at the same drift ratio (except for BMFTP), and this indicated that the prestressing technology had no significant effect on the max state of the specimens. Therefore, the abovementioned results demonstrated that the load–displacement curves were influenced by the section type, and the prestressing technology had beneficial effects on the later stage, after the specimen reached its maximum lateral resistance. In any case, a greater amount of experimental data is needed to support this conclusion.

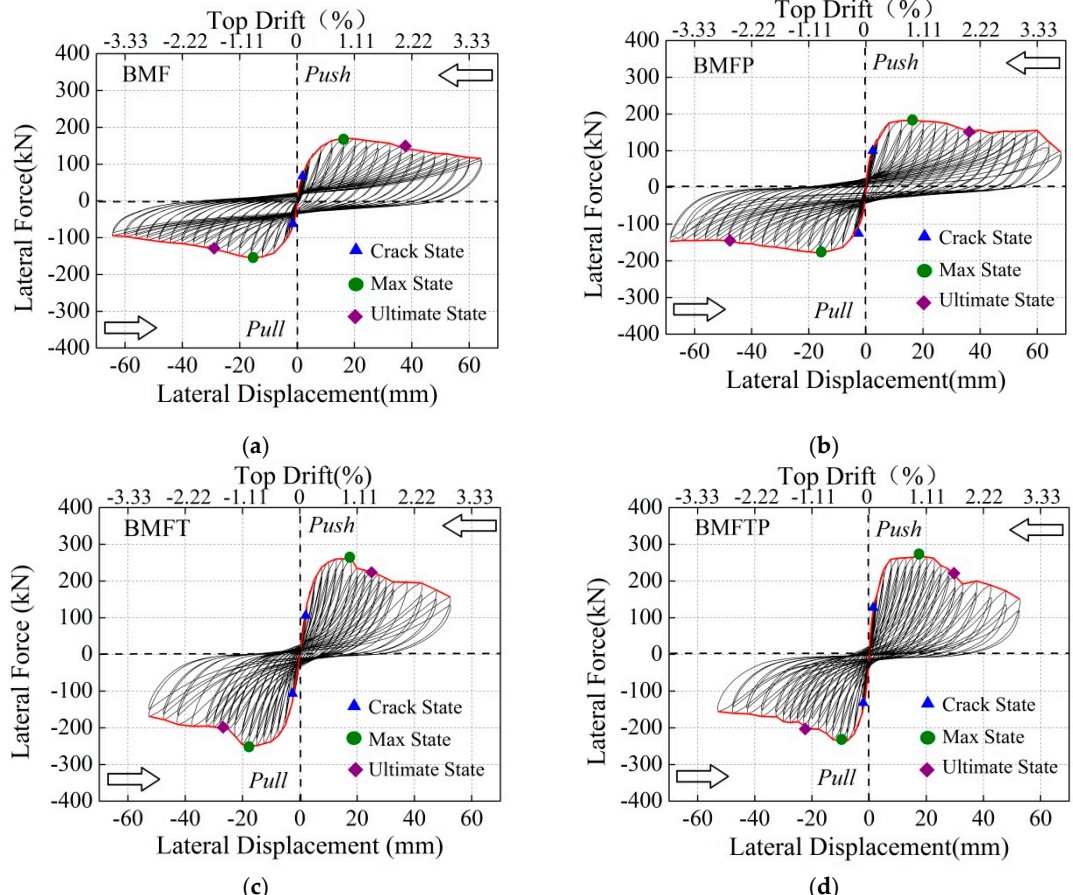

**Figure 7.** Load–displacement curves of the specimens: (**a**) BMF, (**b**) BMFP, (**c**) BMFT, (**d**) BMFTP.

**Table 3.** Summary of displacements and measured loads.

| No. | Direction | Crack State | | | Max State | | | Final State | | |
|---|---|---|---|---|---|---|---|---|---|---|
| | | Q (kN) | △ (mm) | DR (%) | Q (kN) | △ (mm) | DR (%) | Q (kN) | △ (mm) | DR (%) |
| **BMF** | + | +80 | +2.3 | +0.13 | +170 | +16.0 | +0.89 | +115 | +64.0 | +3.55 |
| | − | −70 | −1.5 | −0.09 | −154 | −16.0 | −0.89 | −94 | −64.0 | −3.55 |
| BMFP | + | +100 | +2.2 | +0.12 | +182 | +16.0 | +0.89 | +98 | +68.0 | +3.78 |
| | − | −100 | −2.3 | −0.13 | −178 | −16.0 | −0.89 | −147 | -68.0 | −3.78 |
| BMFT | + | +100 | +1.6 | +0.08 | +262 | +17.5 | +0.97 | +154 | +52.5 | +2.92 |
| | − | −110 | −2.0 | −0.14 | −249 | −17.5 | −0.97 | −168 | −52.5 | −2.92 |
| BMFTP | + | +110 | +1.2 | +0.07 | +268 | +17.5 | +0.97 | +150 | +52.5 | +2.92 |
| | − | -100 | −1.1 | −0.06 | −236 | −10.0 | −0.55 | −150 | −52.5 | −2.92 |

## 4. Discussion of Test Results

### 4.1. Envelope Curves

An envelope curve is made up of the connection of the maximum lateral resistance of each loading cycle, and it is commonly used in the analysis of seismic performance, because it can intuitively reflect the seismic performance characteristics of the specimens. Figure 8a shows the envelope curves of all specimens, and the positive and negative direction envelope curves of the specimens are drawn in the same direction by taking their absolute values, as shown in Figure 8b. Figure 8a shows that the flange in the specimens enhances the lateral resistance of the specimens, and it also increases the initial stiffness of a specimen. Considering the influence of prestressing technology, it could be observed that the prestressing technology improves the initial stiffness of all specimens. Moreover,

the lateral resistance in both the positive and negative direction of the specimen BMFP is higher than that of the specimen BMF at the same displacement, and the curves of the specimen BMFP decline more slowly than those of the BMF specimen, as shown in Figure 8b. A similar situation is observed in BMFT in the positive direction. However, the negative direction comparison results between the BMFT and BMFTP shows that the lateral resistance of BMFTP was lower and reached at the smaller drift than BMFT. This was due to the early occurrence of a severe diagonal crack. In spite of this, the prestressing technology has a beneficial effect on the initial stiffness and declining trend of the lateral capacity of the specimens. Furthermore, the larger plateau on the envelop curves around the max lateral resistance were observed at specimens with prestressing technology than those without, as showed in Figures 7 and 8. This demonstrated that the specimen with prestressing technology can keep the lateral resistance unchanged in a larger displacement range when the specimen reached max lateral resistance, and this also proved that the specimen with prestressing technology had better ductility. Nonetheless, the influence of the prestressing technology on the seismic performance of the specimens is related to the failure mode of the specimens, and a greater amount of supplementary testing is needed to support the analytical conclusions.

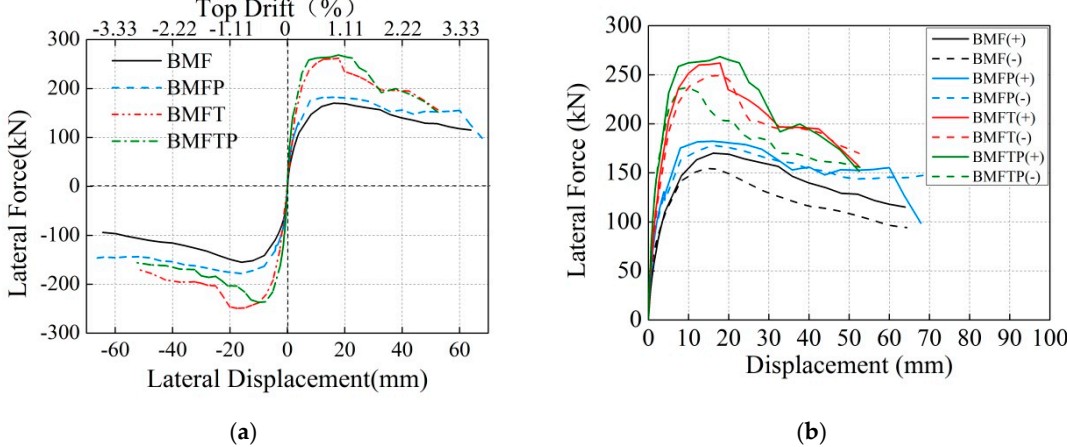

**Figure 8.** Comparison of envelope curves of specimens: (**a**) overall comparison, (**b**) positive and negative direction comparison.

## 4.2. Displacement Ductility

Displacement ductility reflects the deformation capacity of the specimens without significant degradation during the period from the yield state to the ultimate state, and it is related to the failure types of the specimens. There were two states used in calculation of the displacement ductility: the yield state and the ultimate state. However, the yield state of the specimens is not easy to determine because of the complexity of the failure mode and the inconsistency of the computing method. In this study, the energy equivalence method is adopted to calculate the yield state of the specimen, and the calculation method, shown in Figure 9, is also used in the literature [12,27]. Therefore, the displacement ductility $\mu$ of the specimens can be defined as the ratio of the ultimate displacement $\Delta_u$ corresponding to the ultimate state to the yield displacement $\Delta_y$ corresponding to the yield state.

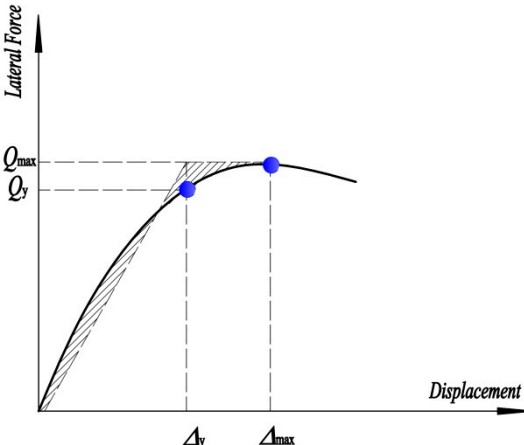

**Figure 9.** Determination of yielding point.

Table 4 provides a summary of the calculation results of the displacement ductility of the specimens. The term $\overline{\mu}$ represents the average values of the displacement ductility $\mu$ in both the positive and negative direction. It can be observed that the displacement ductility varied within a range of values between 3.34 and 6.92, which is clearly related to the section type of the specimens. This can be explained by the different failure modes corresponding to the crack patterns. A flexure and ductility failure occurred in the specimens with a rectangular section, while a shear and brittle failure occurred in the specimens with a T-shape section. With regard to the influence of the prestressing technology, it improves the displacement ductility of the specimens, with an increase of about 49% for the specimens with a rectangular section and 56% for the specimens with a T-shape section. This improvement is also proved by the crack patterns of the specimens. Thin and densely covered cracks occurred in the BMFTP specimen, while several wider main cracks appeared in the BMFT specimen, as shown in Figure 6. This demonstrates that the prestressing technology improves the integrity of the specimens by applying stress to the masonry wall in advance, and it ensures that the specimens have a better deformation potential in terms of their seismic design.

**Table 4.** Summary of calculation results of displacement ductility.

| No. | Direction | Yield State | | | Ultimate State | | | Ductility | |
|---|---|---|---|---|---|---|---|---|---|
| | | Q (kN) | $\triangle_y$ (mm) | DR (%) | Q (kN) | $\triangle_u$ (mm) | DR (%) | $\mu$ | $\overline{\mu}$ |
| **BMF** | + | +143 | +7.7 | +0.43 | +144 | +37.0 | +2.05 | 4.81 | 4.63 |
| | − | −126 | −6.3 | −0.35 | −131 | −28.0 | −1.55 | 4.44 | |
| BMFP | + | +147 | +5.6 | 0.31 | +154 | +35.0 | +1.94 | 6.25 | 6.92 |
| | − | −147 | −6.2 | 0.34 | −151 | −47.0 | −2.61 | 7.58 | |
| BMFT | + | +226 | +6.7 | +0.37 | +222 | +25.5 | +1.42 | 3.79 | 3.34 |
| | − | −226 | −8.3 | −0.46 | −212 | −24.0 | −1.33 | 2.88 | |
| BMFTP | + | +234 | +5.4 | +0.30 | +228 | +28.0 | +1.55 | 5.19 | 5.22 |
| | − | −195 | −4.0 | −0.22 | −201 | -21.0 | −1.17 | 5.25 | |

### 4.3. Stiffness Degradation

Stiffness is an important index of the seismic analysis of structures. It is well known that the stiffness of a specimen decreases with the increase of the displacement [3–7]. With regard to the stiffness degradation, the secant stiffness $K_{si}$ is adopted in this study and calculated by Equation (1). The $V_{\max,i}$ and $V_{\min,i}$ are the peak load resistance of specimen at the $i$ cycle, while the $\Delta_{\max,i}$ and $\Delta_{\min,i}$ are the corresponding displacement at the $i$ cycle, respectively. Moreover, the secant stiffness $K_{si}$ is also normalized by the initial stiffness $K_0$ at the first load cycle to estimate the stiffness degradation, and the relationship between the secant stiffness and drift were plotted in Figure 10.

$$K_{si} = \frac{V_{\text{max},i} - V_{\text{min},i}}{\Delta_{\text{max},i} - \Delta_{\text{min},i}}, \tag{1}$$

The initial stiffness of specimens BMF, BMFP, BMFT and BMFTP is calculated to be 67.3 kN/mm, 109.7 kN/mm, 107.1 kN/mm and 169.8 kN/mm, respectively. It can be observed that the prestressing technology had a significant influence on the initial stiffness of the specimens. An increase in the initial stiffness of 59% is found for the specimens with a rectangular section, while an increase in the initial stiffness of about 55% was found for the specimens with a T-shape section. As shown in Figure 10, it can be clearly observed that the normalized stiffness degradation versus drift curves divided into two categories: the specimens with and without prestressing technology. The specimens with prestressing technology experienced a greater fall in the degradation of the normalized stiffness than the specimens without this technology throughout the loading process. This demonstrates that the prestressing technology had a significant effect on the stiffness degradation of the specimens. On the other hand, the prestressing technology sped up the rate of the normalized stiffness degradation. A similar conclusion is also provided by Hassanli et al. [17], who indicated that the posttension increases the initial stiffness but caused a rapid reduction in the secant stiffness. However, a greater amount of experimental data is needed to support this conclusion.

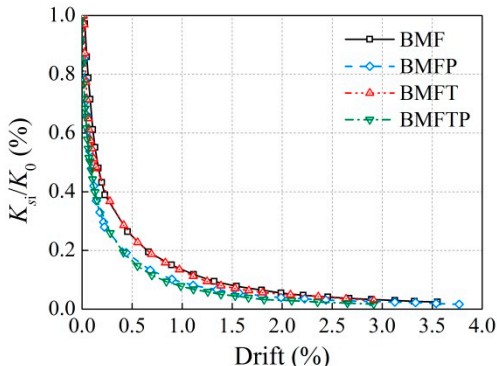

**Figure 10.** Stiffness degradation of the specimens.

## 4.4. Energy Dissipation and Equivalent Viscous Damping

Related to the load–displacement curves, energy dissipation is another important index to evaluate the seismic performance of the specimens. It is generally accepted that the energy dissipation $E_s$ can be measured by the area surrounded by the hysteretic curve in one loading cycle, as shown in the shadowed area of Figure 11. The approach adopted here is also suggested in the literature [12,27–29]. Moreover, in order to evaluate the influence of the prestressing technology more directly, the load–displacement curves are divided into three phases: the ascending stage, max state and descending stage. The comparison displacements in the ascending stage and descending stage are chosen as the first displacement cycle and the displacement when the lateral resistance of the specimens dropped to 85% of the maximum lateral resistance, respectively. It should be noted that the displacement cycle of 10.0 mm is selected for BMFT to be consistent with that of BMFTP, although the negative maximum lateral resistance of the BMFT specimen reached the displacement of 17.5 mm. Table 5 summaries the calculated results of the energy dissipation of the specimens, and the energy dissipation of the specimens versus drift curve is plotted in Figure 12a.

Figure 12a clearly shows that the dissipation energy of the specimens is influenced by the section type of the specimens. The $E_s$ of the specimens with a rectangular section increases with the increment of the displacement, while the $E_s$ of the specimens with a T-shape section increases at the initial stage but remains stable after a drift of 2.5%. For the BMF and BMFP, the comparison results show that the $E_s$ of BMFP is lower than that of BMF at the ascending stage, but the $E_s$ of BMFP is about 15% higher than that of BMF, with an increment of the displacement at the max state and descending stage.

For the BMFT and BMFTP, the comparison results show that the $E_s$ of BMFTP is higher than that of BMFT at the ascending stage and max state, but the $E_s$ of BMFTP is lower than that of BMFT at the descending stage. This indicates that the prestressing technology also improved the energy dissipation of the specimens at the ascending stage. Overall, it demonstrates that the influence of the prestressing technology on the $E_s$ of the specimens is related to the failure mode of the specimens, and it improves the energy dissipation of the specimens before they reach their maximum lateral resistance.

Furthermore, a common and simple indicator to represent the damping of the specimens is the equivalent damping ratio. The equivalent damping ratio $\xi_{eq}$ can be defined as the ratio of the energy dissipation of the specimens in one cycle to the total energy input by the actuator, as shown in Equation (2):

$$\xi_{eq,i} = \frac{E_{s,i}}{2\pi K_{si}\Delta^2_{max,i}}, \tag{2}$$

where $E_{s,i}$ represents the energy dissipation of the specimen in one cycle, $K_{si}$ is the secant stiffness, and $\Delta_{max,i}$ is the maximum displacement in one cycle. Figure 12b shows the relationship between $\xi_{eq}$ versus drift, and the average $\xi_{eq}$ of each specimen is listed in Table 5. Figure 12b clearly shows that $\xi_{eq}$ firstly decreases, but it then increases with the increment of the drift after a drift of 0.2%. Moreover, $\xi_{eq}$ gradually remains constant after the drift reaches 1.0%. The final average values of $\xi_{eq}$ of specimens BMF, BMFP, BMFT and BMFTP are 10.8%, 10.0%, 8.2% and 8.2%, respectively. The $\xi_{eq}$ obtained in this study is similar to that provided in the literature [4]. However, it seems that the prestressing technology used in this study has no marked influence on the equivalent damping ratio of the specimens.

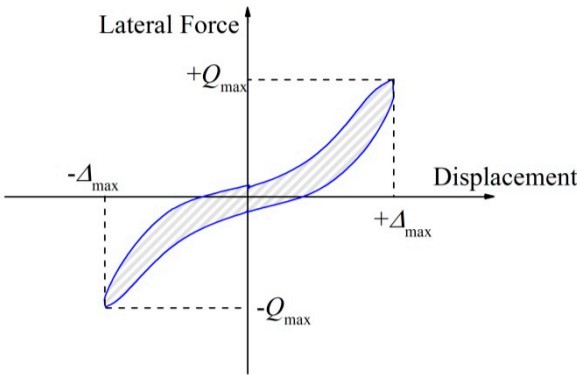

**Figure 11.** Calculation method of energy dissipation.

**Table 5.** Dissipated energy and equivalent viscous damping of all specimens.

| No. | Ascending Stage | | | Max State | | | Descending Stage | | | $\xi_{eq}$ (%) |
|---|---|---|---|---|---|---|---|---|---|---|
| | $\triangle$ (mm) | DR (%) | $E_{s,c}$ (kN/mm) | $\triangle$ (mm) | DR (%) | $E_{s,m}$ (kN/mm) | $\triangle$ (mm) | DR (%) | $E_{s,u}$ (kN/mm) | |
| BMF | 4.0 | 0.22 | 171.6 | 16.0 | 0.89 | 1841.9 | 28.0 | 1.55 | 3381.5 | 10.8 |
| BMFP | 4.0 | 0.22 | 126.6 | 16.0 | 0.89 | 2070.2 | 28.0 | 1.55 | 4024.94 | 10.0 |
| BMFT | 2.5 | 0.14 | 93.9 | 10.0 | 0.56 | 1320.4 | 22.5 | 1.25 | 3133.15 | 8.23 |
| BMFTP | 2.5 | 0.14 | 142.7 | 10.0 | 0.56 | 1402.4 | 22.5 | 1.25 | 2832.45 | 8.19 |

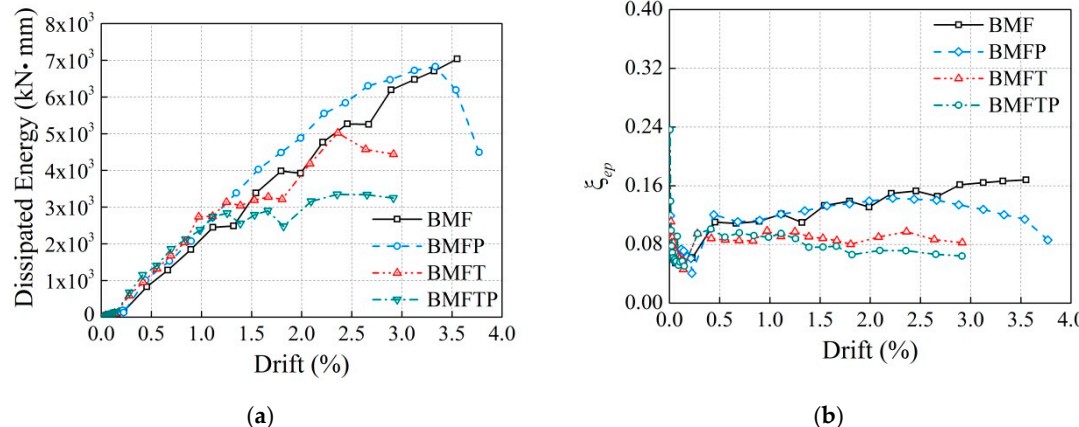

**Figure 12.** Comparison results of the specimens: (**a**) dissipated energy, (**b**) equivalent viscous damping.

## 5. Conclusions

In this study, a new prestressing technology was proposed and used in the reinforced concrete masonry wall systems to improve its seismic performance. Four reinforced concrete block masonry wall systems, consisting of two symmetrically arranged reinforced block masonry walls, with different section types and prestressing technologies, were built and subjected to cyclic load testing. The failure mode and crack pattern of the specimens were presented, and the key parameters, including the displacement ductility, stiffness degradation, energy dissipation and equivalent viscous damping, were discussed. The following conclusions can be drawn:

(1) There were two kinds of failure modes observed in all specimens with different section types. The cracks on specimens BMF and BMFP were concentrated on the top and bottom of the specimens, and few cracks appeared at the middle, while the cracks were distributed uniformly on the face of specimens BMFT and BMFTP, and X-shape cracks occurred at the middle of the wall. This indicated that a flexure and ductile failure occurred in specimens BMF and BMFP, while a shear and brittle failure occurred in specimens BMFT and BMFTP. Moreover, the prestressing technology had no significant effect on the failure state of the specimens, but it influenced the crack propagation, making the cracks fine and densely covered.

(2) A symmetrical and obvious pinching effect was observed in the hysteretic response of all specimens. The specimens with a T-shape section obtained a higher lateral resistance than the specimens with a rectangular section, while a more abrupt and rapid decline was observed in the former in the later stage after the specimens reached their maximum lateral resistance. Moreover, the descending section of the curves of the specimens with prestressing technology are gentler than those of the specimens without this technology, and this indicates that the prestressing technology has a beneficial influence on the descending section of the specimens. Furthermore, the larger plateau on the envelop curves around the max lateral resistance were observed at specimens with prestressing technology than those without, this proved that the specimen with prestressing technology had better ductility on the other hand. However, the influence of the prestressing technology on the lateral resistance of the specimens does not clearly refer to the specimens with a different failure mode, and a greater amount of experimental data is needed to support this conclusion.

(3) The average displacement ductility of the specimens varied within a range of values between 3.34 and 6.92, which is clearly related to the section type of the specimens and failure mode. Moreover, the prestressing technology improves the displacement ductility of the specimens, with an increase of about 49% for the specimens with a rectangular section and 56% for the specimens with a T-shape section. Furthermore, the prestressing technology significantly improved the initial stiffness of the specimens. Additionally, it is found that the specimens with the prestressing technology experienced a greater fall in the degradation of the normalized stiffness than the specimens without this technology throughout the loading process.

(4) It could be seen that the energy dissipation of the specimens is influenced by the section type. The energy dissipation of the specimens with a rectangular section increases with the increment of the displacement, while the energy dissipation of the specimens with a T-shape section increases at the initial stage but remains stable, after drift of 2.5%. In addition, it could be concluded that the prestressing technology improved the energy dissipation of the specimens at the ascending stage. With regard to the equivalent viscous damping, it firstly decreases but increases with the increment of the drift and gradually remains constant after the drift reaches 1.0%. The final average values of specimens BMF, BMFP, BMFT and BMFTP are 10.8%, 10.0%, 8.2% and 8.2%, respectively. However, it seems that the prestressing technology used in this study has no marked influence on the equivalent damping ratio of specimens.

In summary, this paper proposed a new prestressing technology and experimentally investigated the seismic performance of fully grouted concrete masonry walls using new prestressing technology, proving the advantage of new prestressing technology and its potential application in masonry structures. However, the fully grouted concrete masonry walls using new prestressing technology also have brought many new problems, such as higher initial stiffness and faster degradation of stiffness, etc. Therefore, there are still many theoretical and engineering practice problems to be solved. Further experimental and theoretical research still needs be carried out in order to confirm the seismic performance of fully grouted concrete masonry walls using new prestressing technology.

**Author Contributions:** Formal analysis, B.C., Z.Z., and Y.Q.; methodology, F.W.; writing—original draft, B.C.; writing—review and editing, X.Y. and F.W.

**Funding:** This research received no external funding.

**Acknowledgments:** The present work was conducted with the financial support of the National Key R&D Plan-China (Grant No. 2016YFC0701502-3).

**Conflicts of Interest:** The authors declare no conflict of interest.

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
