# Peer review of "Experimental Investigation into the Seismic Performance of Fully Grouted Concrete Masonry Walls Using New Prestressing Technology"

_applsci, doi:10.3390/app9204354_

Round 1
Reviewer 1 Report
The reviewed paper deals with experimental investigation of seismic behaviour of axially prestressed concrete masonry walls or columns.
This topic is very actual and interesting for engineering practice and therefore for scientific research.
Study provides very well prepared and described experimental procedure and specimen. The testing methodology is supported by new scientific research and national regulations.
Also, the paper is well structured, and the authors have clearly emphasized the main contributions and conclusions of the study. The introduction of the study includes a good overview of previous researches.
In the reviewer’s opinion, this paper is good and valuable study. The paper can be accepted for publishing, although the reviewer would have some comments and recommendations for the authors.
The paper deals with short walls which are like the columns. Did the authors think of just one prestressed bar per wall? The cost would be reduced and could have a similar effect of uniform stress. During the cyclic test, did the stresses in the prestressed bars measured and did the reinforcement yielding occur? In the table 2 it would be useful to show the modulus of elasticity and shear strength for masonry prisms if measured. On the page 6, the LVDTs numbering is inappropriate and looks like reference number. It should be corrected. For the experiment, the boundary condition of the upper beam is very important. It is assumed that there was no rotation of the beam during the experiment. Has this been accomplished? in addition to the photo of cracks in Figure 6, it would be convenient to show cracks on the flanges of T section, if any.
The reviewer’s opinion is that the larger plateau on the backbone curves around the maximum shear force value (Figure 7 and 8) shows an advantage of prestressing (higher ductility), but it is not so pronounced that general conclusions can be drawn. Also, an increase the initial stiffness necessarily occurs at prestressing what can generate higher forces on the structure during a real earthquake (especially in the initial stage), which can reduce the positive effect of the prestressing.
Author Response
Dear Reviewer 1,
We appreciate your review and suggestions. They are very valuable and helpful, which are important and necessary for us to modify the manuscript. We have revised our manuscript according to your comments.
Point 1: The paper deals with short walls which are like the columns. Did the authors think of just one prestressed bar per wall? 

Response 1: Thank you for your professional question. In this study, prestressed bars were symmetrically arranged in each wall, and the stress of each bar was different. This was to ensure that the action point of the prestressing force was located at the centroid of section. The position and magnitude of stress on the bars were determined by mechanics calculation. We apologize for the misread and we have added the explanation with this respect at lines 164-166, which make the paper more reasonable.
Point 2: The cost would be reduced and could have a similar effect of uniform stress. During the cyclic test, did the stresses in the prestressed bars measured and did the reinforcement yielding occur?
Response 2: Thank you for your professional comment and question. The object of this paper was the load-bearing walls in one-storey building, and the purpose was to improve the axial stress of the walls by using prestressing technology. This point was the same as that of the reviewer. Much to our regret, we mainly focused on the seismic performance parameters of specimens using the prestressing technology, and we failed to obtain the information of stress variation of reinforcement during the loading process. We appreciate it and we will test the stress variation of reinforcement in our following studies.
Point 3: In the table 2 it would be useful to show the modulus of elasticity and shear strength for masonry prisms if measured.
Response 3: Thank you for your helpful advice. In this study, six masonry prisms were constructed to obtain the material information of fully grouted masonry wall, and they were subjected to axial compression according to Chinese standard for test method of basic mechanics properties of masonry (GB/T 50129-2011). Therefore, the compression strength and modulus of elasticity of masonry prisms were obtained. However, the shear strength of masonry prisms needs to be determined by shear tests, which were not carried out in this study. We appreciate it and have added the modulus of elasticity of masonry prisms in Table 2 to make the paper more valuable.
Point 4: On the page 6, the LVDTs numbering is inappropriate and looks like reference number. It should be corrected.
Response 4: Thank you for your professional advice. We apologise for the misleading of LVDTs numbers. We have revised the corresponding sentence at lines 220-221 and illustrations in Figure 4, which makes the paper more understandable.
Point 5: For the experiment, the boundary condition of the upper beam is very important. It is assumed that there was no rotation of the beam during the experiment. Has this been accomplished?
Response 5: Thank you for your professional comment and question. The boundary condition of the upper beam was determined by the test setup. Two groups of parallel lateral bracings were used at both sides of the specimens to avoid out-of-plane failure, which were located at the reaction frames. Moreover, the horizontal MTS hydraulic actuator was arranged at the centre of the upper beam and they were connected by spherical hinge, which were to ensure the lateral force acting in the centre of the upper beam during the loading process. The above setups were to realize the similarity between the experiments and engineering practice. We are very grateful if the reviewer can accept our explanation and we have revised the sentences at lines 216-218 to make the paper more intelligible.
Point 6: in addition to the photo of cracks in Figure 6, it would be convenient to show cracks on the flanges of T section, if any.
Response 6: Thank you for your helpful advice. We appreciate it and have added the cracks photos of flanges of T section in Figure 6e and 6f, which makes the paper more valuable.
Point 7: The reviewer’s opinion is that the larger plateau on the backbone curves around the maximum shear force value (Figure 7 and 8) shows an advantage of prestressing (higher ductility), but it is not so pronounced that general conclusions can be drawn.
Response 7: Thank you for your professional and insightful comment. We discussed the effect of prestressing on initial stiffness and max lateral resistance of specimens in the section of envelope curves, and the ductility of specimens were discussed in a separate section which were determined by calculation. We appreciate your professional comment, and we added the corresponding sentences at lines 337-342 in Section 4.1 and lines 476-478 in Conclusion, which makes the paper more intelligible.
Point 8: Also, an increase the initial stiffness necessarily occurs at prestressing what can generate higher forces on the structure during a real earthquake (especially in the initial stage), which can reduce the positive effect of the prestressing.
Response 8: Thank you for your professional comment. In this research program, the advantages of the fully grouted concrete masonry wall with prestressing technology were evaluated in two aspects: the seismic performance and the practical application. According to the cyclic test results, the seismic performance of the specimens with and without prestressing technology was discussed. We found that the prestressing technology had no significant effect on the failure mode of the specimens, but it influenced the crack propagation. Moreover, the prestressing technology improved the ductility of specimens and energy dissipation, but it had no marked influence on the equivalent damping ratio. With regard to the stiffness, it can be concluded the prestressing technology increased the initial stiffness of specimen. We agree with the reviewer opinion that an increase the initial stiffness necessarily occurs at prestressing what can generate higher forces on the structure during a real earthquake. The purpose of this study is to improve the seismic performance of fully grouted concrete masonry walls, and prestressing technology is to increase the normal stress of the walls actually. Therefore, the specimen with prestressing technology can be regarded as the specimen under additional axial stress, and the stiffness of specimen with prestressing technology can be determined by equivalent method and considered in advance in engineering design. Therefore, the effect of prestressing technology on the stiffness of specimen can be solved.
We appreciate for your professional comment and we are grateful if the reviewer can accept this explanation. We have modified the conclusion and further work of the study in the revised manuscript to make the paper more understandable (see lines 504-507).

Reviewer 2 Report
In the study, a seismic strengthening approach to masonry structures using prestressing technology is adopted, and the relevant construction method is described in detail. The coupled masonry shear walls are selected as the research subject, and the seismic performances of the coupled masonry shear walls are studied using a cyclic test.
General comment:
In the reviewer's opinion the paper is generally well-written. The literature review regarding the numerous experimental and theoretical studies on various aspects of masonry structures, with the aim of improving their seismic performance, is very extensive and comprehensive. Nevertheless, the reviewer believes that the text of the paper is somehow lacking in some important and crucial information on its novelty/originality.
The main problem is that it is hard to understand what the novelty of the paper is as it was not clearly stated. The authors stated that “This study proposes a new prestressing technology to improve the seismic performance of masonry structures”. However, in the manuscript (in Introduction) only the literature review, as well as the information about the content of the paper, are provided. The novelty of the research was not enough emphasized, i.e. the ‘new prestressing technology’ was not highlighted and explained sufficiently in the manuscript.
The authors need to decide whether the “the new prestressing technology” is (1) about the wall systems, subjected to cyclic force, that consisted of two symmetrically arranged reinforced block masonry walls or (2) two categories of speciments in the experiment program: the rectangular section category and T-shape section category or (3) the magnitude of the prestressing and the application method of the prestressing. I think the main problem is that the paper is presenting precisely planned and executed experiments but failing to demonstrate a novelty/originality of them.
Therefore the paper in its present form is not suitable for publication. Consistent information about the novelty of the research (in the Introduction as well as in conclusion) has to be added. The paper should be revised as suggested and re-submitted.
Specific comments:
When improving your manuscript, please:
(1) expand the abbreviations: BMF, BMFP (line 133), BMFT and BMFTP (line 139) and LVDT (line 205),
(2) change the indications [1], [2]and [3] in lines 205-210, since [x] are usually reserved for references.
Author Response
Dear Reviewer 2,
We appreciate your review and suggestions. They are very comprehensive and insightful, which are important and necessary for us to modify the manuscript. We have revised our manuscript according to your comments.
Point 1: The main problem is that it is hard to understand what the novelty of the paper is as it was not clearly stated. The authors stated that “This study proposes a new prestressing technology to improve the seismic performance of masonry structures”. However, in the manuscript (in Introduction) only the literature review, as well as the information about the content of the paper, are provided. The novelty of the research was not enough emphasized, i.e. the ‘new prestressing technology’ was not highlighted and explained sufficiently in the manuscript.
The authors need to decide whether the “the new prestressing technology” is (1) about the wall systems, subjected to cyclic force, that consisted of two symmetrically arranged reinforced block masonry walls or (2) two categories of speciments in the experiment program: the rectangular section category and T-shape section category or (3) the magnitude of the prestressing and the application method of the prestressing. I think the main problem is that the paper is presenting precisely planned and executed experiments but failing to demonstrate a novelty/originality of them.
Response 1: Thank you for your professional comments. We are very sorry for not expressing clearly the novelty of this study, and we are grateful if the reviewer can accept this explanation and revised manuscript.
The background of this study was that the seismic performance of low-stories masonry structures in rural areas was poor. In the section of introduction, we firstly summarized the methods to improve the seismic performance of masonry structure. The methods can be simply divided into two types: new structural form and new strengthening technology. Secondly, we discussed the several prestressing technology used at present and the influence of prestressing technology on the seismic performance of masonry walls. Thirdly, we concluded that the interaction relationship between the walls and diaphragm cannot be neglected by summarizing the previous tests. Finally, we proposed a new prestressing technology on the masonry wall system and carried out an experimental study.
With regard to the problem mentioned by reviewer, it was noted that “the new prestrssing technology” named in this study referred to a new construction method of prestressing technology. This new construction method used the vertical holes in the concrete masonry units and vertical bars with threads to apply prestressing force. The prestressing bars were bonded in grouted concrete, which is the difference from the previous unbounded prestressed masonry. The magnitude of prestressing and the application method of the prestressing were described in detail in this study. Therefore, the new method was the main innovation of this study. In addition, the fully grouted concrete masonry wall system was the test object of this study, and the two section types of rectangular section and T-shape section were the test variables of this study. The above two aspects were chosen to study the influence of new prestressing technology on the specimens. In other words, the above two aspects contributed to the study of new prestressing technology.
In summary, this study proposed a new prestressing technology to improve the seismic performance of masonry structures, and the influence of the new prestressing technology on the specimens were tested and discussed. We apologize for not being able to express the novelty of this study, and we have modified the corresponding sentences in the sections of Abstract, Introduction and Conclusion of the study in the revised manuscript to make the paper more understandable (see lines 16-19, lines 113-123 and lines 452-455).
Point 2: expand the abbreviations: BMF, BMFP (line 133), BMFT and BMFTP (line 139) and LVDT (line 205)
Response 2: Thank you for your helpful comment. We apologise for the misunderstanding of abbreviations in this study. In this study, the abbreviations BMF, BMFP, BMFT and BMFTP were used for naming specimens, which were listed in Table 1. The specimens consisted of two symmetrically arranged reinforced block masonry walls, and its shape similar to frame. Therefore, the abbreviations BM represented the bare masonry walls, and the abbreviations F and FT represented the masonry walls without and with T-shape section, respectively. Moreover, the abbreviation P represented the prestressing technology. In addition, the LVDT (line 205) represented the linear variable differential transformers which were used to measure the deformation of specimens. Considering the reviewer’s valuable comment, we have added the explanations for the abbreviations BMF, BMFP, BMFT and BMFTP in lines 140-144 and for LVDT in lines 220-221, which makes the paper more intelligible.
Point 3: change the indications [1], [2]and [3] in lines 205-210, since [x] are usually reserved for references.
Response 3: Thank you for your professional opinion. The indication in lines 205-210 originally used to facilitate the marking the LVDTs at different positions. We apologise for the misuse of indications. We have revised the corresponding sentence at lines 221-225 and illustrations in Figure 4, which makes the paper more comprehensible.

Round 2
Reviewer 2 Report
In the new version the novelty/originality of the paper is more clearly stated. The authors expressed the novelty of their study, and modified the corresponding sentences in the sections of Abstract, Introduction and Conclusion of the study. In the revised manuscript the main goal of the paper is understandable. The authors in the revised version of the paper claryfied what is the novelty of the study.
Other minor suggestions have also been considered (i.e.: lines 220-221, lines 221-225 and illustrations in Figure 4).
Hence, I suggest that the paper can be accepted in the present form.